# Risky Roads in Kuwait: An Uneven Toll on Migrant Workers

**DOI:** 10.3390/ijerph19159726

**Published:** 2022-08-07

**Authors:** Janvier Gasana, Soad Albahar, Mahareb Alkhalidi, Qout Al-Mekhled, Darline El Reda, Marwan Al-Sharbati

**Affiliations:** 1Department of Environmental and Occupational Health, College of Public Health, Kuwait University, P.O. Box 24923, Kuwait City 13110, Kuwait; 2Department of Community Medicine and Behavioral Sciences, College of Medicine, Kuwait University, P.O. Box 24923, Kuwait City 13110, Kuwait; 3Department of Social and Behavioral Sciences, College of Public Health, Kuwait University, P.O. Box 24923, Kuwait City 13110, Kuwait; 4Division of Public Health, Michigan State University, East Lansing, MI 48824, USA

**Keywords:** road safety, risky driving, traffic citations, road traffic accidents, road traffic mortality, Kuwait

## Abstract

This descriptive study reveals trends in citations and traffic-related mortality in Kuwait. Secondary data were utilized, where data on road traffic citations were obtained from the traffic police in the Ministry of Interior for the years from 2011 to 2015, and road traffic mortality data for the study period were obtained from the Ministry of Health. Objective: To describe recent trends in data related to road traffic safety in Kuwait over time, which could serve as an important indicator for the level of enforcement of existing traffic regulations. Descriptive summary statistics are presented. Results: There was a total of 24.2 million traffic violations during the study period. The number rose dramatically from 4 million citations in 2011 to nearly 6.5 million in 2015. The indirect method of citation (issued indirectly via surveillance methods) constituted a higher percentage of citations, 70.4%, compared to the direct method of citation (issued directly by the police officer), 29.6%. Furthermore, the top reason for citation was speeding, followed by parking in no parking/handicapped zones, driving with an expired license, and crossing a red light. Road traffic fatalities (RTFs) in Kuwait from 2011 to 2015 totaled 2282. About 450 people die each year in Kuwait from road traffic injuries and a slightly decreasing trend was found. Non-Kuwaitis have RTF counts that are four times higher than Kuwaitis, with 1663 and 263 deaths, respectively. Conclusions: Road traffic safety continues to be a major problem in Kuwait. Increases in citation issuance show a rise in traffic regulation enforcement, yet risky driving behaviors continue to account for most violations issued. Harsher penalties, road safety education, and implementing graduated driving licensing may be warranted to increase the safety of the roads.

## 1. Introduction

Road traffic injuries (RTIs) are listed as the 12th leading cause of death globally, across all age groups, and were responsible for approximately 1.28 million deaths every year as of 2019 [1,2]. Road traffic accidents (RTAs) are considered a major cause of mortality, disability, and economic loss worldwide [1,3]. A country’s income level is strongly associated with the risk of road traffic death. Despite lower rates of vehicle ownership, low-income countries experience death rates due to road traffic injuries that are three times higher than high-income countries [1,4]. Contrary to this finding, however, several higher-income countries in the Middle East experience traffic-related death rates that are unusually elevated for high-income countries [5]. In recent years, Kuwait has experienced rapid development and increasing vehicular growth, which has resulted in a substantial increase in road accidents and loss of life, particularly among young adults [6]. RTAs accounted for 64.4% of the 4886 reported accidental deaths referred for medico-legal examination in Kuwait from 2003 to 2009 [7]. Ziyab and Akhtar [5] showed that from 2000 to 2009, 11,591 non-fatal RTIs occurred in Kuwait, of which 28.2% were severe, and 3891 were fatal RTIs. Most of the RTI-related deaths were found to be amongst males (87.3%) in the age range of 20–59 years (70.8%) [5]. As of 2019, road injuries in Kuwait are ranked as the second leading cause of death and disability-adjusted life years (DALYS) in the country [8].

Several studies have been conducted to elucidate the determinants of these markedly higher road traffic fatality rates among high income countries in the Gulf Cooperation Council (GCC). Lack of seat belt use, driving at excess speed, inattention, and incorrect vehicle maneuvering were considered to be major factors in the occurrence of accidents which resulted in death or injuries [9,10,11]. Studies from Kuwait reported that road traffic accidents were mostly caused by lack of adherence to existing traffic rules and regulations, driver carelessness, and relaxed licensing procedures [12]. Kuwait has implemented several strategies over the last decade focused on improving road traffic safety, such as the enactment of a seat belt law, implementation of a penalty point system, increased investment in surveillance cameras, and participation in the World Health Organization’s Decade of Road Traffic Safety [13,14,15]. Despite this, studies from Kuwait reported that human factors, such as careless driving behaviors, are the main cause of traffic collisions in Kuwait, yet enforcement of traffic regulations in the country is very low [16].

Migrant workers take a hit when it comes to injury and mortality for many reasons, including the different driving behaviors and driving licensing procedures in their countries of origin and the low level of enforcement of road traffic regulations in Kuwait [16,17]. Most migrant workers occupy professions that do not match their professional profile and often work longer hours in risky jobs in the production, labor, and service sectors, when compared to local workers [18,19]. Based on service occupations, non-Kuwaiti males hold taxi, delivery, or public transport driver jobs that could be considered as particularly risky. In 2017, Akarametagul [17] showed that the causes of migrant workers road accidents in Thailand were carelessness and not knowing traffic rules and regulations such as traffic light and warning signs. Urgent and effective actions are needed to make roads safer for Kuwaitis and non-Kuwaitis.

The objective of this study is to summarize and describe traffic citations and traffic-related mortality in Kuwait from 2011 to 2015. By examining data on the citations issued over time, the study identifies the main determinants of road traffic accidents in Kuwait. The study also aims to assess the relationship between citations and traffic-related mortality and proposes context-specific recommendations for Kuwait to reduce mortality based on the Haddon matrix, which is widely used worldwide for the prevention of road traffic injuries.

## 2. Methods

### Data and Analysis

This study used secondary data which were sourced from the Ministry of Interior (Department of Statistics, Division of Traffic) for the years 2011 to 2015. Mortality data due to road traffic collisions were obtained from The Ministry of Health for the study period. A descriptive data analysis and statistical tests were conducted. From the citation data, we summarized the number of violations, gender of the driver, and method of citation (direct vs. indirect). A direct citation is one that is issued by a police officer when the driver is present, and an indirect citation is one that is issued via surveillance camera or by a police officer when the driver is not present. We also summarized the type of offenses that warranted citations over the years and ranked them to determine the top 10 reasons for traffic violations.

Mortality data included the following variables: number of deaths, age, gender, nationality, and year. Age was divided into 8 categories by increments of 10 years: 0–10, 11–20, 21–30, 31–40, 41–50, 51–60, 61+, and unknown. In the dataset, nationality was divided into multiple categories including: ‘Kuwaiti’, ‘GCC country’, ‘Arab’, ‘Asian’, ‘Other’, and ‘unknown’. We summarized these labels into 2 categories: ‘Kuwaiti’ and ‘non-Kuwaiti’, with the latter encompassing all labels apart from ‘Kuwaiti’. We then classified traffic-related fatalities for each of the variables: age, nationality, and gender. Analyses were carried out using SPSS statistical software version 28.0.1.1 (IBM Corp., Armonk, NY, USA).

## 3. Results

### 3.1. Citations

There was a total of 24.2 million traffic violations during the study period in Kuwait. Figure 1 summarizes the number of traffic citations from 2011 to 2015 in total and by gender. Overall, the number rose dramatically from 4 million citations in 2011 to nearly 6.5 million in 2015, with a sudden surge occurring after 2013. The figure also shows that males had a larger number of citations than females throughout the years, with males accounting for about 72% of citations.

The total number of traffic citations were also stratified by method of issuance (direct vs. indirect), as shown in Table 1. The direct method of citation (issued directly by the police officer) constituted 29.6% of all the citations, while the indirect method of citation (issued indirectly via surveillance methods or when the driver is not present) constituted 70.4% of all the citations. There was also a marked increase in the number of indirect citations being issued in 2014 and 2015 with a corresponding decrease in direct citations. When stratifying by gender, females received fewer citations from both direct and indirect measures, as expected. Male drivers held more indirect citations (67.5%) than their female counterparts (20%). Similarly, males had higher direct citations compared to females, 82% and 18%, respectively. A larger disparity is shown in direct citations, with males having 4.5 times more direct citations than females. Company-leased cars, on the other hand, accounted for only 8.8% of the traffic citations overall—more specifically, 0% direct and 12.5% indirect citations.

The top 10 reasons for traffic citations for the study period have been summarized in Table 2. Speeding is the top reason and held the number one slot for the studied 5 years. It is followed by parking in a no parking/handicapped zone, driving with an expired license, and crossing a red light. The year 2015 accounted for more than 20% of citations over the study period. The number one reason for citation that year was exceeding the speed limit, with over 3.5 million citations issued, representing nearly 55% of all the citations in 2015. The top three reasons alone (exceeding the speed limit, parking in designated handicap spaces, and parking in no-parking or waiting zones) accounted for nearly 80% of all citations.

### 3.2. Mortality

From 2011 to 2015, there were a total of 2282 documented deaths resulting from RTAs in Kuwait. Figure 2 shows the total number of traffic-related fatalities over the 5 years. On average, about 450 people die each year in Kuwait because of a road traffic injury. A decreasing trend is seen throughout the years, with 2015 showing 64 less deaths than 2011.

In terms of gender differences, overall, males and females account for 54% and 46% of traffic-related deaths, respectively. There have been fluctuations in the number of deaths over the years. Males and females show an alternating pattern of who takes the lead in death counts. In 2011, males accounted for 99.8% of the traffic-related deaths; in 2012, 92.5% of the deaths were females; and in 2013, about 83% of deaths were males. This alternating pattern continued until 2015, when the death counts balanced out.

Figure 3 summarizes the total road traffic deaths in Kuwait by age. The age groups that consistently had the highest number of traffic-related deaths were the 21–30 and 31–40 groups (24.9% and 19.7% of deaths, respectively), followed by those aged 11–20 years (17.6%) and 41–50 years (15.5%). The number of RTFs stratified by nationality from 2011 to 2015 is shown in Table 3. Non-Kuwaitis had traffic-related death counts that were four times higher than Kuwaitis (1663 [72.9%] and 263 [11.5%] deaths, respectively). Those that are of unknown nationality are seen to represent 15.6% of deaths.

## 4. Discussion

### 4.1. Enforcement of Regulations

The enforcement of existing traffic regulations has increased over time, as evidenced by increases in citations issued (for drivers of all vehicles including passenger vehicles, vehicles for heavy goods, and other vehicles) over the study period. Speeding was the number one reason for violations, accounting for a great majority of all citations issued for the studied 5 years and 55% of all citations issued in 2015 alone. Overall, RTF rates have been decreasing slightly over time.

### 4.2. Citations

Over the 5 years, there was an increase in citations, and the surge of increase seemed to occur after 2013, whereafter the citations doubled. Overall, there were more indirect citations issued than direct ones. The increase in the number of documented traffic violations might have occurred due to the investment in traffic surveillance technology, since citations from the indirect method increased dramatically, whereas the citations from the direct method decreased. This shows that the cameras on the roads serve their intended purpose.

Males in general had a larger number of citations than females throughout the years. This is expected, as there is ample evidence in the scientific literature that males are more likely to engage in risk-taking behaviors, such as speeding, road racing, crossing red lights, driving before having a license, and driving under risky conditions [20,21]. According to Koushki and Bustan [22], among the young population in Kuwait, females are usually safer drivers than their young male counterparts, who smoke while driving, use seatbelts less, and have a higher rate of traffic accidents. More specifically, males show a higher percentage of indirect citations than females. It is important to note that indirect citations are issued to the Civil ID to whom the car is registered. Some cars may be registered to the female drivers’ fathers or brothers. Therefore, some of the percentages may be inflated, especially with more citations being issued via indirect methods such as surveillance cameras. An even larger disparity is shown in the direct method, where males have a 4.5 times higher count of citations than females. It is likely that there are missed opportunities for the enforcement of traffic regulations among female drivers in Kuwait. Company-leased cars account for fewer citations than individual-leased cars. These cars are usually driven by non-Kuwaiti males working for private and/or public companies in Kuwait.

In line with previous research on the most common causes of road traffic accidents in the GCC and in Kuwait, we found that the top reason for citations was exceeding the speed limit [9,10,11,12]. Parking in no parking or handicap zones, having an expired driver’s license, and bypassing red lights were the next most common causes of citations issued in Kuwait. This shows how drivers are careless and do not adhere to existing traffic rules and regulations. One common dangerous behavior among drivers in Kuwait that is not included within the reasons for citations is tailgating, or not keeping a safe distance from the preceding car [23,24]. A study by Al-Hemoud et al. [16] found that 85% of drivers in Kuwait keep less than two car-length following distances on freeways. Studies in Kuwait have shown that rear-end crashes account for 30.4% of road accidents [25]. Tailgating is clearly a major cause of road accidents and is not considered to be a reason for citation when it should be, especially in Kuwait, where speeding is prominent and where weather conditions such as dust storms and fog can reduce visibility on roads and contribute to road accidents [26].

### 4.3. Mortality

There was a slowly decreasing trend in deaths each year in Kuwait because of RTIs. This may be due to the adherence of seat belt use as proposed by Koushki et al. [13]. Overall, males and females accounted for roughly half of traffic-related death counts, respectively. It is important to note that these numbers do not reflect that the population is growing over time. Rates of deaths per 100,000 population are a better measure of risk than number of deaths. Mortality rates allow us to consider the size of the population and compare our performance to those of other countries. According to the World Health Organization, as of 2015, road injuries accounted for 24.5 deaths per 100,000 persons in Kuwait, decreasing significantly from 26.7 per 100,000 persons in 2010 [27]. Overall, RTF rates in Kuwait have been decreasing over time, yet remain higher than expected as compared to other high-income countries such as the USA and UK, where the road injury mortality rates in 2015 were 10.9 and 2.6 per 100,000 persons, respectively [27].

The death rates stratified by gender show that males in Kuwait have a death rate that is 4.7 times larger than females [27]. This is similar to the global trend where male mortality rates exceed that of females. According to data from the U.S. Department of Transportation’s Fatality Analysis Reporting System (FARS), many more men than women die each year in motor vehicle crashes [28]. Men typically drive more miles than women and are more likely to engage in risky driving practices, including not using safety belts, driving while impaired by drug and/or alcohol, and speeding. Furthermore, crashes involving male drivers often are more severe than those involving female drivers [29]. According to Hamza et al. [30], 83% of road traffic accidents in Bahrain were the fault of male drivers and only 14% were females, potentially explaining the higher traffic-related death rates.

As for age, the highest traffic-related deaths are seen in the 21–30 and 31–40 age groups, followed by the 11–20-year category. This is in line with research suggesting that young drivers between the ages of 20 and 30 cause more RTAs [31]. Furthermore, young people contribute to a higher road traffic accident mortality rate compared to other age groups and have slightly higher odds of incurring an injury from these accidents [30,32,33]. In the United Kingdom, although drivers aged 17–19 only make up 1.5% of UK license holders, they are involved in 12% of serious and fatal crashes [34]. This could be because the younger age groups have less driving experience and are more likely to commit violations in driving from negligent or careless behavior [31]. Another reason for increased road traffic accidents and related fatalities in the younger age group is that youth in GCC countries consider car driving as entertainment and tend to engage in speeding or joyriding for amusement [10].

Non-Kuwaitis showed higher counts of traffic-related deaths than Kuwaitis throughout the study period. Non-Kuwaitis have higher counts of death because they account for 70% of the 4.2-million-person population of Kuwait [35]. Much of the non-Kuwaiti population is young, mostly male migrant workers [36]. Therefore, considering the size of each of these populations is crucial to accurately compare between groups. There are also potential reporting errors in the data, specifically in nationality categorization. Different sub-categories for nationality have been included and the data did not specify whether these categories were mutually exclusive. For example, the ‘GCC country’ category and ‘Arab’ category could overlap, and persons could have been classified twice, as they fit both categories, which would result in inflated non-Kuwaiti death counts.

### 4.4. Conceptualization of Potential Strategies

Instead of enforcing traffic regulations through the issuance of citations, there is a need for alternative or additional strategies aimed at impacting driver behavior to avert RTAs and their subsequent injuries and fatalities. Lack of progress in injury control stems from the failure to recognize that injuries cannot occur without the action of specific agents such as vehicles in this case. This is analogous to how infectious diseases are transmitted through vectors [37]. The interplay between the agent and the characteristics of the host and the environment constitute the classic epidemiological triad that determines injury distribution, which is not random. According to the Haddon matrix, modifiable factors that contribute to traffic injury could be classified into human, vehicle, and environmental factors [38]. All these factors are potential targets for preventive interventions. Modification of these factors depends heavily on understanding the socioeconomic and political context in which they operate [39]. As identified in the Haddon Matrix when applied to motor vehicle injuries, most pre-crash prevention can be attributed to the vehicle driver’s attitudes and actions [40,41].

The study adds to existing evidence that driver carelessness and risky driving behaviors leading to violation of traffic regulations are prominent among the population of Kuwait [16,31]. Tackling road traffic injuries with a focus on ‘People’ and ‘Pre-event’ will probably prevent a lot of injuries in Kuwait, since most can be attributed to improper human behavior and driver irresponsibility. The first area of development is road safety education. Health education agencies—for example, local community health centers—could be involved in increasing awareness for safe driving. We are also in need of improved and stricter driver and vehicle licensing and testing procedures to ensure that drivers are experienced. Another important suggestion is to encourage the use of mass transport. However, such action necessitates improving this underutilized sector. Education measures should also target policymakers and the police force, in order to address the low levels of enforcement of traffic regulations.

With regards to enforcement, Luca [42] found that driver behavior actually responds well to traffic tickets, showing that police officers are an important component of road safety policy. Exploring the impact of harsher penalties for speeding and for repeat violators could be promising. Furthermore, the increased enforcement of penalties for engaging in known risky behaviors such as lack of seat belt use, children in improper restraints, and the use of mobile phones while driving should be considered to reduce traffic-related fatalities. Other possible approaches to reducing fatalities include allocating more resources toward municipalities with higher population densities and increasing traffic enforcement at night, since tickets have a larger impact during nighttime. Furthermore, Graduated Driver Licensing (GDL) systems, that are currently used by many countries, would be deemed as beneficial to implement in Kuwait to combat the lax licensing procedures and reduce the burden of young driver crashes. This licensing procedure requires young drivers to go through stages of varying restrictions, such as adult supervision, daytime driving, and passenger limits, before achieving full licensure [43]. These GDL provisions allow new, young drivers to enhance their driving skills in low-risk settings to combat driving behaviors known to substantially increase their fatal crash risk [44,45]. Migrant workers who are to drive in Kuwait need to receive training as soon as they arrive. The training should familiarize these workers with traffic laws and regulations, warning signs, traffic lights, symbols that are used on the road, penalties and dangers from violations and negligence, and information on what should be done in the case of an accident [17].

Citation issuance coupled with road safety education prevents drivers from adopting reckless driving habits such as speeding, running red lights, and failing to fasten their seat belt [46]. This will undoubtedly contribute to minimizing RTAs and RTFs.

### 4.5. Strengths and Limitations

To our knowledge, this is the first study to examine data on the citations issued in Kuwait over time, which can serve as an important indicator for the level of enforcement of existing traffic regulations. The study identifies the main determinants of RTAs based on the most common reasons for traffic citations issued. In doing so, it assesses the effects of Kuwait’s implementation of strategies to overcome the traffic-related issue from 2011 to 2015. The study also assesses the relationship between citations and traffic-related mortality and proposes context-specific recommendations for Kuwait to reduce mortality based on the Haddon matrix. There are some limitations, however, including the possibility of the inflation of male citations due to automatic issuance to the Civil ID to whom the car is registered for indirect citations, and cultural factors for direct citations, where females are less likely to be approached in person. There are also potential reporting errors in the data, specifically in nationality categorization, as previously mentioned. Moreover, there was no information on which countries were included for each of the nationality categorizations; only broad categories such as ‘Asian’ and ‘Arab’ were used. Underreporting and disorganized data are major limitations in road traffic data in general. Improved classification and recording of traffic violations is of utmost importance, as there is no uniform data collection method or forecasting system in place in Kuwait [15]. Finally, the study only explored fatal RTI’s, excluding non-fatal RTI’s due to a lack of data.

## 5. Conclusions

Our findings indicate that overall road traffic fatality rates have slightly decreased over time, yet they remain higher than expected as compared to other “high-income” countries, where road safety has been taken seriously, and preventive measures are implemented appropriately. The enforcement of existing traffic regulations has increased over time, as evidenced by an increase in the number of citations issued, yet risky driving behaviors such as speeding and crossing red lights continued to account for the great majority of all violations issued for the studied 5 years. This suggests that harsher penalties may be warranted to increase the safety of the roads in Kuwait. There is also a need for implementing road safety education programs as well as establishing GDL systems for young drivers to reduce collision rates. Furthermore, since there is a higher risk of road traffic-related mortality in non-Kuwaitis, most of whom are migrant workers, training for this subset of the population with a focus on one rectangle (People and Pre-event) of the Haddon Matrix is essential.

We recommend further research to explore the variations of traffic citations and fatalities by month. Studies in Saudi Arabia have shown that road traffic accidents increase during the months of December and May, as well as during Ramadan [10,47,48]. Knowing such information can help with resource allocation to tackle the issue more efficiently. It would also be useful to identify how different types of fines would impact the volume of RTAs in the country over time.

Another suggestion is to analyze individual-level citation data linked to Civil ID to identify the demographic characteristics of violators. Future qualitative research is recommended to explore the reasons for violators’ actions and their consideration for other road users in Kuwait. Moreover, future research is important because it will help the researchers to understand why and how Kuwaiti citizens have developed fear and anxiety towards using the roads due to the volume of RTAs. Another great way to address the issue of road traffic is the use of social media campaigns, internet forums, and education at the school level. Finally, implementing a system for data exchange between the traffic police, road and transport agencies, hospitals, and fire and ambulance services in Kuwait needs to be considered [15]. Such an interface would contribute to further traffic safety research through supplying data that can be used to assess and forecast trends in RTAs.

## Figures and Tables

**Figure 1 ijerph-19-09726-f001:**
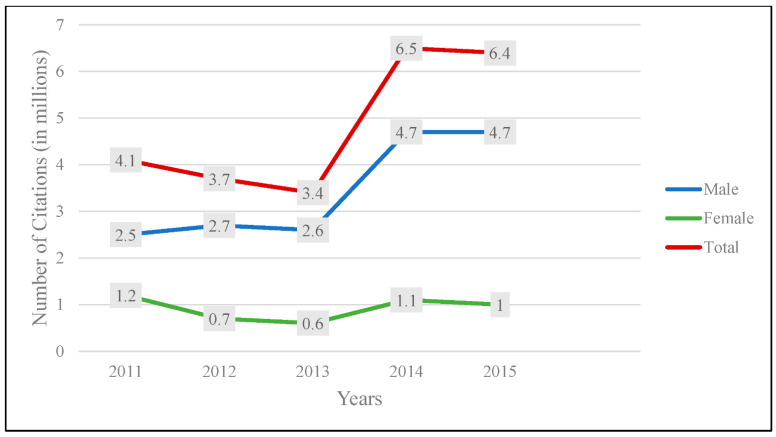
Total number of traffic citations by gender from 2011 to 2015.

**Figure 2 ijerph-19-09726-f002:**
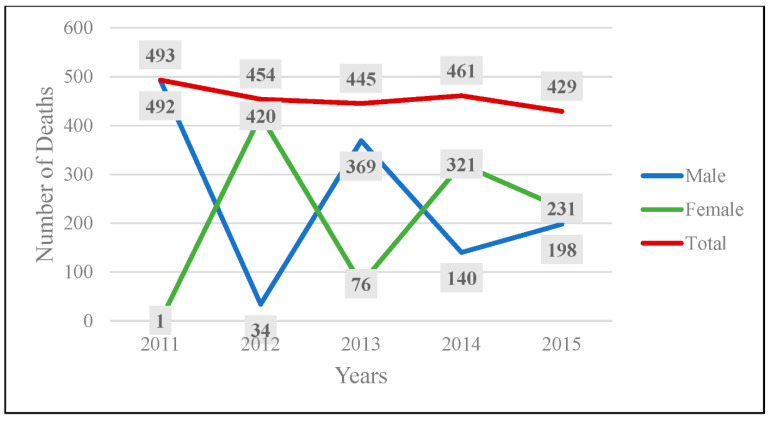
Total deaths resulting from road traffic accidents by gender in Kuwait from 2011 to 2015.

**Figure 3 ijerph-19-09726-f003:**
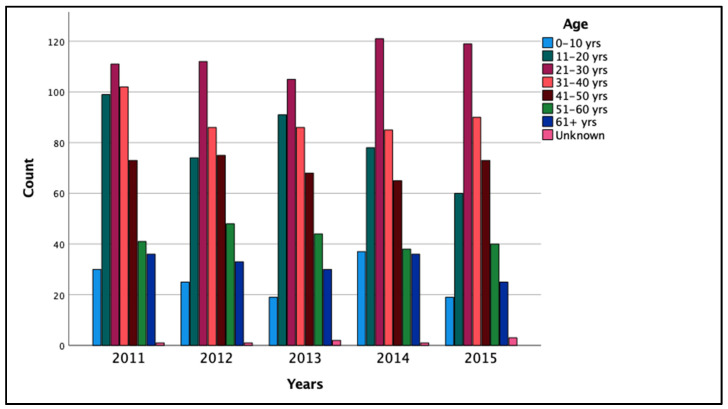
Total road traffic deaths in Kuwait by age from 2011 to 2015.

**Table 1 ijerph-19-09726-t001:** Percent of traffic citations per year by method of issuance (direct vs. indirect) from 2011 to 2015.

*Year*			*Type of Citation*	
			*Direct*	*Indirect*
	2011	*Percent of citations*	43.5%	56.5%
	2012		31.8%	68.2%
	2013		42.9%	57.1%
	2014		18.1%	81.9%
	2015		24.1%	75.9%
*Total*			29.6%	70.4%

**Table 2 ijerph-19-09726-t002:** Top 10 Reasons for Traffic Citations: 2011–2015.

*Reason for Citation*	*Frequency*	*Percent (%)*
1. *Exceeding the Speed Limit*	11,371,348	46.8
2. *Parking in No Parking/Waiting Zones*	3,213,568	13.2
3. *Parking in Designated Handicap Spaces*	2,148,524	8.8
4. *Non-renewal of Driver’s License or Permit*	2,015,420	8.3
5. *Bypassing Red Light*	1,986,911	8.2
6. *Violation of Security and Quality Conditions*	702,453	2.9
7. *Tinting Vehicle Windows to Amounts not Permitted*	506,929	2.1
8. *Others*	436,564	1.8
9. *Not Wearing Seat Belt*	302,319	1.2
10. *Driving without Driver’s License*	187,275	0.8

**Table 3 ijerph-19-09726-t003:** Total number of traffic-related deaths by nationality from 2011 to 2015.

*Years*			*Nationality*			*Total*
			*Kuwaiti*	*Non-Kuwaiti*	*Unknown*	
	2011	*Count*	31	426	36	493
		*Percent within year*	6.3%	86.4%	7.3%	
	2012	*Count*	33	373	48	454
		*Percent within year*	7.3%	82.2%	10.6%	
	2013	*Count*	44	333	68	445
		*Percent within year*	9.9%	74.8%	15.3%	
	2014	*Count*	65	311	85	461
		*Percent within year*	14.1%	67.5%	18.4%	
	2015	*Count*	90	220	119	429
		*Percent within year*	21.0%	51.3%	27.7%	
*Total*		*Count*	263	1663	356	2282
		*Percent total*	11.5%	72.9%	15.6%	

## Data Availability

The datasets were obtained from the Ministries of Interior and Health. Requests for data access may be made to these Ministries.

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
