# Peer review of "Risky Roads in Kuwait: An Uneven Toll on Migrant Workers"

_ijerph, 2022, doi:10.3390/ijerph19159726_

Round 1
Reviewer 1 Report
The authors addressed a very important topic related to road safety. I am always curious about this issue especially in other countries. However, I have a few comments:
- the numbering of the chapters and subchapters should be improved as they are all numbered 1,
- the dates of access to websites are not given in the bibliography,
- furthermore, the discussion does not say whether the data were for drivers of passenger vehicles, heavy goods vehicles or all vehicles,
- training should of course be included in future research, but also social campaigns, internet forums, education at school level,
- Furthermore, nothing is known about the length of time a person has held a driving licence and this is also related to experience,
- aggression, which is one of the causes of accidents in, for example, European countries, is also not taken into account, as is mobile phone use,
- the age of the vehicle and its technical condition could be taken into account in further studies, because, as the authors write, it is not only a human factor, but also the environment, the environment and the technical factor of the vehicle,
- the state of the roads, it may be worth comparing the results of research not only to the USA and the UK, but also to neighbouring countries, to see how safety looks there and how efforts are made to improve it
Thank you!
Author Response
Responses to Reviewer 1
- The numbering of the chapters and subchapters should be improved as they are all numbered 1,
Response: Thank you for noticing this. We just corrected the numbering.
- The dates of access to websites are not given in the bibliography,
Response: We just included the dates of access to the websites. Thanks again for this.
- Furthermore, the discussion does not say whether the data were for drivers of passenger vehicles, heavy goods, vehicles or all vehicles,
Response: We clarified in the discussion that the data was collected for drivers of all vehicles (passenger, heavy goods, and others).
- Training should of course be included in future research, but also social campaigns, internet forums, education at school level,
Response: We just added that important suggestion in the discussion section.
- Furthermore, nothing is known about the length of time a person has held a driving licence and this is also related to experience,
Response: This is a great observation, but the data did not have the information.
- Aggression, which is one of the causes of accidents in, for example, European countries, is also not taken into account, as is mobile phone use,
Response: These are very valid points, but the data did not have that information.
- The age of the vehicle and its technical condition could be taken into account in further studies, because, as the authors write, it is not only a human factor, but also the environment, the environment and the technical factor of the vehicle,
Response: The suggestion is noted, and it will be taken into consideration when conducting further research studies in Kuwait. Also, another thing worth noting is that Kuwait removed all old cars from circulation.
- The state of the roads, it may be worth comparing the results of research not only to the USA and the UK, but also to neighbouring countries, to see how safety looks there and how efforts are made to improve it
Response: Given our experience driving in Kuwait, the roads are in good shape, but the volume of vehicles is huge for those roads. However, the lanes of roads in the USA are much wider. Regarding the situation in neighboring countries, the data is scarce but is being built up. In future research, it will be possible to compare the roads in Kuwait with the roads in the neighboring Gulf countries.

Reviewer 2 Report
Introduction - There was mention of Thailand as a comparison. However, other developed country such as the USA, UK and Canada would have been useful to cite mainly of the volume of vehicles on those roads and what are the their levels of RTAs. This comparison would have been useful considering the increase of vehicles on Kuwait's roads.
Methods - There were no mention of qualitative or quantitative approach. Also, was the sample representative of all Kuwait or a specific region?
Recommendation for Future Research - This should have been after the conclusion as it is the final part of the research. Would this study recommend future qualitative research to understand the reasons for violators actions and their consideration for other road users in Kuwait? Is future research important to understand of Kuwaiti citizens have developed fear and anxiety towards using the roads due to the volume of RTAs? Would be useful to conduct future research on how different types of punishments would impact on the volume of RTAs?
Word Choice Corrections
Line 58/59 - occurrence of accidents which resulted in death or injuries.
Line 65 - Risky, replace with dangerous or careless.
Line 72 - Tend to, replace with often.
Line 72 - Risky jobs. Do you mean jobs that involves fatigue?
Line 73 - of the, replace with based on the.
Line 88 - sentence construction, replace with, This study used secondary data which was sourced from the Ministry of.
References
There are some inconsistencies with some references missing dates.
Author Response
Responses to Reviewer 2
Introduction - There was mention of Thailand as a comparison. However, other developed country such as the USA, UK and Canada would have been useful to cite mainly of the volume of vehicles on those roads and what are the their levels of RTAs. This comparison would have been useful considering the increase of vehicles on Kuwait's roads.
Response: We did not compare the roads in Kuwait with the roads in USA, UK, and Canada because of the difference between the state of the roads between Kuwait and those countries. The volume of vehicles is big but the roads in Kuwait can’t not keep up with it and they don’t have many lanes as it is the case, at least in USA and Canada where we drove. So, Thailand was the closest for the comparison.
Methods - There were no mention of qualitative or quantitative approach. Also, was the sample representative of all Kuwait or a specific region?
Response: Well, we only use the data that was handed to us by the Ministries of Interior and Health. I would say we used quantitative approach since we have that data. We can’t talk of the sample representativeness given the fact that we were given the only readily available data. Kuwait does not have a consistent system to collect the data. We are working with the both Ministries of Interior and Health to help them improve the collection of the data.
Recommendation for Future Research - This should have been after the conclusion as it is the final part of the research. Would this study recommend future qualitative research to understand the reasons for violators actions and their consideration for other road users in Kuwait? Is future research important to understand of Kuwaiti citizens have developed fear and anxiety towards using the roads due to the volume of RTAs? Would be useful to conduct future research on how different types of punishments would impact on the volume of RTAs?
Response: We addressed this by removing the recommendation for future research to the conclusion section. We also included the great recommendations you outlined above and thank for that.
Word Choice Corrections
Line 58/59 - occurrence of accidents which resulted in death or injuries.
Response: We included the correction in the text.
Line 65 - Risky, replace with dangerous or careless.
Response: We replaced risky with careless. Thank you.
Line 72 - Tend to, replace with often.
Response: We replaced Tend to with often.
Line 72 - Risky jobs. Do you mean jobs that involves fatigue?
Response: We mean jobs that are associated with a lot of injuries.
Line 73 - of the, replace with based on the.
Response: We replaced the “Of the”, with “based on the”, and we edited the sentence a little bit.
Line 88 - sentence construction, replace with, This study used secondary data which was sourced from the Ministry of.
Response: We replaced the sentence with the sentence that you suggested above.
References
There are some inconsistencies with some references missing dates.
Response: We added the dates for access of websites and we fixed all the inconsistencies.
